# Health-Related Quality of Life Assessment in Older Patients with Type 1 and Type 2 Diabetes

**DOI:** 10.3390/healthcare11152154

**Published:** 2023-07-28

**Authors:** Špela Volčanšek, Mojca Lunder, Andrej Janež

**Affiliations:** 1Clinical Department of Endocrinology, Diabetes and Metabolic Diseases, University Medical Centre Ljubljana, Zaloška 7, 1000 Ljubljana, Slovenia; mojca.lunder@kclj.si (M.L.); andrej.janez@kclj.si (A.J.); 2Medical Faculty, University of Ljubljana, Vrazov trg 2, 1000 Ljubljana, Slovenia

**Keywords:** patient-reported outcome measures (PROMs), self-rated health, aging, HbA1c, diabetes complications, obesity prevention, quality in healthcare

## Abstract

Type 1 (T1D) and type 2 diabetes (T2D) are determinants of health-related outcomes including health-related quality of life (HRQOL). We aimed to determine differences in HRQOL between older adults with T1D and T2D and specific factors influencing HRQOL in this age group. This study used a cross-sectional design with 56 age- and HbA1c-matched T1D and T2D patients (aged 68.9 ± 7.8 years; 55% had T2D). We employed several validated questionnaires (Short Form-36 (SF-36) and the EuroQol-5 Dimensions/Visual Analog Scale (VAS)) to investigate the relationships between HRQOL domains and diabetes type, glycemic control, complications, and comorbidities. T1D was associated with better self-reported general health (assessed with the SF-36 general health domain (*p* = 0.048) and the EuroQol-5 VAS (*p* = 0.002), whereas no significant differences in the other SF-36 domains, self-reported diabetes distress, anxiety, or depression were found. Most HRQOL domains were not associated with HbA1c or the presence of diabetes complications. The most significant reduction in HRQOL was experienced by patients with higher BMIs, irrespective of the diabetes type. The obtained HRQOL data could be used in clinical settings for evidence-based patient education focused on specific subgroups of patients, as well as in national healthcare policies, e.g., interventions designed to alleviate obesity.

## 1. Introduction

Diabetes is a common and widespread chronic disease in older adults [1]. According to the National Institute of Public Health and European Health Interview Survey (EHIS), the estimated total number of people diagnosed with diabetes in Slovenia is 145,200, which accounts for the 8.9% prevalence of diabetes in adults. According to survey data in the year 2019, our prevalence of diabetes (of both types) was as follows for the listed age groups—65–74 years: 18.2%, 75–84 years: 20.8%, and 85+ years: 19.2% [2]. However, the majority of elderly diabetes patients have type 2 diabetes (T2D) [1]; therefore, older type 1 (T1D) patients are scarcely represented in the literature. On the other hand, due to improvements in medical care and technology, there are now millions of adults living worldwide with T1D who were diagnosed in their childhood. In addition, adult onset of T1D is not uncommon. In 2022, 329,000 (62%) of all newly diagnosed T1D cases worldwide occurred in people aged 20 years or older [1]. The International Diabetes Federation (IDF) Europe Region, which our country accounts for, has the highest number of individuals with T1D [1]. In Slovenia, 1774 of the total of 6559 T1D individuals are estimated to be aged above 60 years. This number still accounts for only 1.8% of the elderly diabetes patient population (nearly 100,000) in our country [1].

Both T1D as well as T2D have detrimental effects on various health aspects including health-related quality of life (HRQOL) [3,4], with 24 healthy years of life lost per person with T1D [1]. HRQOL independently and more accurately predicts mortality in diabetic patients than some physiological diabetes-specific factors [5,6,7,8,9,10,11]. HRQOL is a composite of physical- and emotional-health-encompassing symptoms, functional status, and longevity, which are dependent on comorbidities [12,13]. Self-rated mental health is viewed as an important domain of HRQOL since the Center for Disease Control and Prevention (CDC) defines HRQOL as an individual’s perceived physical and mental health over time [14]. The multidomain concept of HRQOL is difficult to measure; however, medical science has developed several generic (assessing specific domains) and disease-specific instruments, which are most frequently used in the form of validated questionnaires. The most widely used generic questionnaires, e.g., the EuroQol-5 Dimensions/Visual Analog Scale (EQ-5D/VAS) Questionnaire [15,16,17] and Short Form-36 (SF-36) [18,19], can capture diabetes-related health burdens as well [20]. In addition, several instruments specifically measure diabetes distress, e.g., the Problem Areas in Diabetes (PAID) scale [21]. The Hospital Anxiety and Depression Scale (HADS) questionnaire is used in many different patient or general population settings to capture the symptoms and severities of anxiety and depressive disorders [22].

Overall, HRQOL in T2D and T1D patients is worse than in the general population [23,24,25,26,27]. However, these findings are not consistent across all age groups; e.g., a systematic review revealed no differences in QOL domains between the pediatric population with T1DM and healthy controls [28]. The literature consistently identifies sociodemographic determinants such as female gender, rural environment, lower income, lower education level, and lifestyle choices such as physical inactivity and smoking to be associated with poorer HRQOL in both diabetes types [29,30,31,32,33,34,35,36]. The recommended lifestyle adjustments and complex pharmacological treatment schemes for diabetic patients may affect their QOL from the outset. Furthermore, requiring insulin treatment and the presence of diabetes complications are generally associated with worse HRQOL [37,38,39,40,41,42,43,44]. Some researchers have proven associations between certain domains of HRQOL and glycemic control [30,45,46,47,48,49]; however, most researchers have shown that after adjusting for other factors, HbA1c is not associated with HRQOL [50,51]. Age is another complex determinant of self-rated health in any chronic condition since some authors have concluded that the impacts of many influencing factors decrease with age. Consequently, certain HRQOL domains converge in the oldest age groups in individuals with or without diabetes [52,53]. Furthermore, various comorbidities (such as depression, obesity, dyslipidemia, and hypertension) impact health status in T2D as well as in the general population [54,55,56,57,58]. Body mass index (BMI) is predictive of lower HRQOL in T2D, but evidence for this association in T1D patients is scarce or lacking [32,59,60]. 

HRQOL is an important and understudied topic in diabetes, especially in older adults with multimorbidity [57,61,62]. Comparisons between diabetes types have rarely been investigated, especially in age-matched subjects [63]. Identifying the factors influencing HRQOL is of critical research and social interest [64] since psychosocial care is dictated by guidelines [65,66].

The present study aimed to estimate HRQOL and self-rated mental health in older adults with T1D and T2D to determine both differences between the diabetes types as well as the associations of HRQOL domains with diabetes control and chronic complications.

## 2. Materials and Methods

The present study used a cross-sectional cohort design. Diabetes patients were recruited at their regular outpatient clinic visits at the Department of Endocrinology, Diabetes, and Metabolic Diseases of the University Medical Centre Ljubljana between 1 June and 30 November 2019. Patients eligible for inclusion were T1D or insulin-treated T2D patients aged 60 years or more, using multiple daily injections of insulin (MDI). To ensure greater homogeneity, we only included T2D patients with insulin treatment regimens using at least four applications of insulin daily and T1D patients undergoing MDI therapy and not using any advanced diabetes technology for insulin delivery or sensing (see [67] for further details of the study protocol). The major exclusion criterion was having severe comorbidities of nondiabetic etiology that could profoundly impact general health (such as debilitating musculoskeletal disease, advanced heart failure, etc.) With this measure, the researchers aimed to exclude the possible impacts of other comorbidities on the results. The reasons for the non-inclusion of the remaining patients were the following: refusal to participate, severe visual impairment that made participants unable to fill in the questionnaires, use of insulin pumps or insulin sensors (T1D), premixed insulin therapy use (T2D), and non-insulin injectable therapy use (T2D). Non-ambulatory patients of both diabetes types were not included in the cohort because they did not attend the outpatient department where the participant recruitment was carried out. The major obstacle to patient recruitment was an unanticipatedly low number of T1D candidates. According to the number of patients attending regular check-ups at our Outpatient department, maximally 90 T1D patients were expected to be eligible for inclusion in our study. The study population consisted of 56 of the originally included 60 patients. Patients that did not fill in all the questionnaires (*n* = 4) were excluded.

Ethical approval was obtained from the Slovenian National Ethical Committee (number 0120-703/2017/3). All subjects were informed of the aims of this study, which was conducted according to the World Medical Association Declaration of Helsinki. Written informed consent was obtained from every included patient before entering this study.

Upon inclusion, several medical and anthropometric data were reviewed including diabetes duration, diabetes treatment regimen, the presence of chronic diabetic complications (microvascular and macrovascular), and other comorbidities (arterial hypertension and obesity). Each patient’s weight and height were measured, and BMI was calculated by dividing weight (measured in kilograms) by height (measured in meters) squared [68]. HbA1c was determined with a blood sample using a high-performance liquid chromatography analyzer (D-100, Bio-Rad Laboratories, Inc., Hercules CA, USA). 


**Questionnaires**


Several generic HRQOL questionnaires were used: Short Form-36 (SF-36) [69], EuroQol-5 Dimension (EQ-5D) [17], EuroQol-VAS (EQ-VAS) [70], the Hospital Anxiety and Depression Scale (HADS) [22,71], and the diabetes-specific Problem Areas in Diabetes (PAID) [21] scale. The questionnaires were validated in the Slovenian language. The participants had unlimited time to fill in the required questionnaire forms. The study population consisted of 56 of the originally-included 60 patients. Patients that did not fill in all questionnaires (*n* = 4) were excluded.

### 2.1. Medical Outcomes Short Form-36 (SF-36)

SF-36 is a widely used generic instrument for measuring HRQOL with international validity and reliability. It encompasses 36 questions divided into 8 health dimensions: physical functioning (PF), social functioning, (SF), role limitation due to physical functioning (LPH), bodily pain, mental health (MH), role limitation due to emotional problems (LEH), vitality, and general health perception (GH). Each item is assessed from 0 (worst well-being) to 100 (best well-being); therefore, higher scores indicate higher quality of life [69].

### 2.2. The EuroQol-5 Dimension

The EQ-5D is a standardized generic measure of health for clinical and research use developed by the EuroQoL Group. The EQ-5D comprises the following 5 dimensions: mobility, self-care, usual activities, pain/discomfort, and anxiety/depression. Each dimension has 3 levels: no problems, some problems, and severe problems. The respondent indicates his/her health state in the box next to the most appropriate statement for each of the 5 dimensions. This results in a 1-digit number expressing the level selected for that dimension [17].

### 2.3. The EQ Visual Analogue Scale

The EQ-VAS is a vertical visual analog scale that retains values between 0 (worst imaginable health) and 100 (best imaginable health) on which patients provide global assessments of health [70].

### 2.4. The Hospital Anxiety and Depression Scale (HADS)

The HADS is a self-assessment questionnaire that was developed to be a reliable instrument for detecting depression and anxiety in an outpatient clinical setting. The questionnaire comprises seven questions for anxiety and seven questions for depression, which are scored separately. Cut-off scores are available for quantification; for example, a score of 8 or more for anxiety has a specificity of 0.78 and a sensitivity of 0.9, and for depression, a specificity of 0.79 and a sensitivity of 0.83 [22,71].

### 2.5. The Problem Areas in Diabetes (PAID) Scale

PAID is a simple self-reported 20-item questionnaire for detecting diabetes-related distress with high scientific validity, and, to our knowledge, it is the only diabetes-specific HRQOL questionnaire that has been validated in the Slovenian language. PAID scores are generated by participants indicating the degree to which each of the 20 items is a problem for them from 0 (not a problem) to 4 (a serious problem). The scores are multiplied by 1.25 and summarized to a total score out of 100 [21].

## 3. Statistical Analysis

Descriptive statistics were used to present basic demographics and disease-related data. The Kolmogorov–Smirnov test was used to determine the normality of distribution. Continuous data were presented as means ± standard deviation (SD) and percentages and numerals for categorical variables. An independent sample t-test was used to compare the means of normally distributed variables.

Multivariable linear regression was used to identify the factors that most affected respondents’ self-rated health scores (outcome variables, dependent variables) including diabetes type (T2D or T1D), the presence of microvascular and macrovascular complications, diabetes duration, basic demographics (gender and age), and BMI (independent variables), and these were adjusted for potential confounders. Assumptions of a normal distribution of a dependent variable (using the Shapiro–Wilk test), a linear relationship between an independent and dependent variable, and the absence of multicollinearity between the independent variables were checked for. The linear regression models provided standardized beta coefficients for each group. Statistical significance was set at *p* < 0.05. 

Statistical analyses were performed using IBM SPSS Statistics, version 19 (IBM Corp, Armonk, NY, USA).

## 4. Results

### 4.1. Baseline Patient Characteristics

The participants’ mean age was 68.9 ± 7.8 years, with a mean HbA1c value of 7.9 ± 1.5, mean BMI of 29.8 ± 7.2, and diabetes duration of 27.4 ± 15.5 years, and 55% of participants were male. All participants were exclusively insulin-treated using intensive insulin regimens. The T1D and T2D groups were numerically balanced and age- and HbA1c- matched. No significant differences in the prevalence of microvascular and macrovascular complications, as well as of arterial hypertension, were present (Table 1). As expected, the T1D patients had longer diabetes durations and a lower mean BMI.

### 4.2. HRQOL Outcome Measures

#### 4.2.1. Medical Outcomes Short Form-36 (SF-36)—Health Dimensions

A comparison between the diabetes types revealed higher scores reported by T1D patients in most domains; however, only the general health perception domain demonstrated significantly different scores, denoting better self-rated health (Table 2). The health status domains of physical functioning (PF), role limitation due to physical functioning (LPH), role limitation due to emotional health (LEH), vitality, mental health (MH), social functioning (SF), and bodily pain did not significantly differ between the groups.

#### 4.2.2. EQ-VAS State of Health

The mean EQ-5D VAS score was significantly lower in T2D patients (61.6 ± 17.2 vs. 77.5 ± 16.2; *p* = 0.002), indicating a worse overall impression of self-rated health compared with what was found in T1D patients (presented in Figure 1).

#### 4.2.3. EQ-5D Dimensions

The examination of each of the 5 dimensions of the EQ-5D—mobility, self-care (1.7 ± 0.2 vs. 1.3 ± 0.1; *p* = 0.1), usual activities (2.2 ± 0.3 vs. 1.6 ± 0.2; *p* = 0.06), pain/discomfort (2.2 ± 0.2 vs. 1.8 ± 0.2; *p* = 0.2), and anxiety/depression (1.6 ± 0.2 vs. 1.4 ± 0.2; *p* = 0.3)—showed higher (worse) rating scores for T2D patients, among which only the mobility domain was significantly worse (2.4 ± 0.3 vs. 1.6 ± 0.2; *p* = 0.038), indicating more problems with mobility in the everyday lives of T2D patients (see Figure 2).

#### 4.2.4. The Hospital Anxiety and Depression Scale (HADS) and the Problem Areas in Diabetes (PAID) Scale

The mean HADS A (5.6 ± 2.1 vs. 4.3 ± 3.0; *p* = 0.9) and HADS D (4.1 ± 2.9 vs. 4.2 ± 02.9; *p* = 0.3) scores did not predict high probabilities of anxiety or depression in any of the diabetes types and did not significantly differ between the diabetes types. The PAID scores were relatively low as well (18.7 ± 13.9 vs. 15.8 ± 12.8; *p* = 0.4), indicating low levels of diabetes distress that were not significantly different between the diabetes types (see Figure 3).

#### 4.2.5. Predictors (Diabetes Type, Presence of Diabetes Complications, Diabetes Duration, Basic Demographics, and BMI) Affecting Respondents’ Self-Rated Health Scores

Participant gender, age, diabetes duration, glycemic control, and the presence of diabetes complications had no significant negative impact on either score. T1D had a positive impact on EQ-5D-VAS scores. BMI was negatively associated with the SF-36 physical functioning, bodily pain, general health, and vitality domains, while the remaining domains (SF-LPH, SF-MH, and SF-LEH) and the results of the PAID, HADS A, and HADS D questionnaires were not significantly different between the T1D and T2D participants in our study.

The impacts of diabetes complications on the SF-36 domains and EQ-5D, EQ-5D-VAS, PAID, and HADS scores are presented with a multivariate linear regression analysis in Table 3.

## 5. Discussion

In this cross-sectional study, we examined differences in HRQOL and self-rated mental health in older adults with diabetes according to diabetes type. T1D patients assessed their general health more positively compared with age-matched T2D patients. Furthermore, while exploring the factors impacting the QOL, most of the HRQOL domains were not associated with HbA1c or the presence of diabetes complications. However, the multiple regression model suggests that BMI is a predictor of self-rated health irrespective of diabetes type.

Diabetes is a chronic, progressive, and debilitating disease with multiple complex impacts on various health aspects [72,73]. In addition to glycemic control variables, patient-reported outcomes are gaining recognition as separate outcome measures since many new treatment options are arising and the patient-centered approach is increasingly favored [74,75]. This study used both validated disease-specific instruments as well as patient-reported measures for outcomes that address health-status-related QOL irrespective of disease type (the SF36, EQ-5D, and EQ-VAS questionnaires). Our results indicate better self-rated health in T1D patients compared with age-matched T2D patients, which has been previously described by other authors [20,76]; however, most previous findings on the differences between these two groups have been inconclusive. Only insulin-treated T2D patients were included to ensure better homogeneity among the two groups; therefore, we hypothesize that this difference can be attributed to the treatment regimen [77]. The included insulin-treated T2D patients underwent many treatment intensifications during the disease, and treatment intensification can harm the quality of life of patients [73,78,79]. Some studies elucidated higher scores in T2D individuals, especially when noninsulin therapy was used [37] and even when the T2D individuals were insulin-treated [80]. However, some researchers found no differences in HRQOL between diabetes types in a cohort of adults aged 18 years and older [32]. The literature on the differences in QOL determinants between the T1D and T2D groups is generally scarce. To our knowledge, this is one of the first studies conducted in a subgroup of older age-matched patients. 

Self-rated mental health was evaluated using the HADS questionnaires, one domain of EQ-5D, and the diabetes distress tool PAID. Regardless of relatively low distress and anxiety scores, no differences in self-reported distress, anxiety, or depression were proven between T1D and T2D participants in this study. It is known that both anxiety and depression, which commonly coexist, are poorly recognized by clinicians and that diabetic patients are more prone to these diseases due to the burden of diabetes self-management [29,52,81,82]. To explain the low scores, we hypothesize that long diabetes durations and higher age contribute to the perception of disease burden and that older adults become accustomed to disease burden. In line with our findings, convergence in the oldest age group in subjects with and without diabetes has been described in the literature [52]. A similar phenomenon of a low prevalence of depression and emotional distress is observed in long-standing T1D [83].

The multiple regression model showed no impact of glycemic control on HRQOL or self-rated mental health. Our findings follow the results of other studies that determined that HbA1c was not negatively associated with HRQOL after adjusting for other factors [50,51]. Studies that specifically investigated glycemic control found that increases in HbA1c in the years following diagnosis were independently associated with increased odds of reporting a negative impact of diabetes on QOL [84]. Furthermore, some authors have established that the lower the HbA1c is, the better the diabetes-specific HRQOL is, especially in younger individuals with T1D [46]. Conversely, our cohort was diametrically different concerning the age and duration of diabetes.

QOL determinants in adults with T1D were not impacted by the presence of micro- or macrovascular complications in our cohort, which is an unexpected finding that is not supported by the previous literature [43]. The occurrence or presence of late complications in diabetes was shown to harm HRQOL in the landmark UKPDS study [85] and ADVANCE study [38] in both T2D patients as well as in T1D patients in a DCCT/EDIC follow-up [40]. Diabetic complications with the largest disutility (blindness and amputation) affect QOL more pronouncedly [38]. Accordingly, it was demonstrated that T2D patients without complications had only slightly lower HRQOL than their similarly aged counterparts without diabetes [51]. Furthermore, since HRQOL measures patients’ perceptions of the effects of disease, the presence of different diabetes complications further significantly stratifies patients, and microvascular complications appear to have a more negative impact than macrovascular complications [39]. In general, age powerfully impacts physical and psychological aspects as well; consequently, a phenomenon of older adults accepting their age-related health problems has been described, and the subjective assessment of problems diminishes with older age in some findings [53], which could explain our findings.

The results of multiple linear regression for patient characteristics concerning aspects of quality of life measured using the SF-36 and EQ-VAS questionnaires showed that a higher BMI predicts a lower quality of life, irrespective of the diabetes type. This finding proved to be significant in physical but not mental well-being. This agrees with the findings of other authors investigating both T2D patients as well as other chronic conditions, which determined that physical health worsened as BMI increased [32,56,59,60]. On the contrary, this notion was not proven in a mixed T1D and T2D cohort [20]. Based on these findings, preventing weight gain should be a priority, regardless of the patient’s diabetes type or age, since obesity is a controllable comorbidity with important health implications. We believe this is the most important conclusion of our study since preventative measures can be employed to target obesity, in both healthcare policies as well as on an individual level, through guided, individualized, and structured patient education programs. Furthermore, the presence of obesity and a higher average BMI could explain the differences in HRQOL between the diabetes types in certain physical domains of HRQOL.

The major limitation of our study was its small sample size and the known impacts of this on data analysis. Due to insufficient data available for adult populations, even in the most recent editions of the International Diabetes Federation (IDF) Atlas, the number of people living with T1D across the globe has only been estimated for children and not for the elderly [1]. In addition, the prevalence estimate for those less than 20 years of age did not take into account potential changes in incidence and mortality over time [1]. Evidently, the numbers of T1D patients decline with high age, studied herein, due to mortality. Importantly, the development of the Type 1 Diabetes Index (T1Dndex) has only recently, in the year 2022, offered estimates to be calculated for all ages across different countries [1]. The relatively low estimated national number of older adults with T1D (1.8%) within the total diabetes population in this age group did not allow for more rigorous inclusion. Although the relatively small number of participants should be noted, the number was balanced between T1D and T2D patients. For known-group validity, there were significant differences only in diabetes duration and average BMI variables, which cannot be matched acknowledging the different etiologies of T1D and T2D. Because of the cross-sectional design of our study, the associations do not guarantee causality, and it would be preferable to retest the results in a longitudinally designed setting [86]. The prevalence of complications was high following long diabetes durations; however, we did not further stratify the severity of diabetes complications. This research was conducted in an outpatient setting, wherein the majority of patients are ambulatory, indicating that the complications were non-debilitating and confirming the presence of some selection bias. Several instruments for measuring outcomes were used that could have presumably led to interviewee exhaustion; however, patients were given unlimited time to complete them.

The main strength of this study is the use of validated and reliable instruments for measuring patient-reported outcome measures (PROMs). Given the complex nature of the interaction of diabetes-related factors with HRQOL, PROMs and glycemic control are central outcomes in clinical diabetes care, but they should be evaluated separately [3]. The generic instrument Short Form-36 (SF-36), used in this study’s setting, has been widely and repeatedly used with diabetes patients [49]. The performances of EQ-VAS [15] and EQ-5D have been extensively tested for reporting PROMs across many chronic diseases [20,70]. Furthermore, EQ-5D has been suggested to be a predictor of mortality and first hospitalization in the elderly [87]. The HADS was found to perform well in assessing the symptoms and severity of anxiety disorders and depression in many different patient settings [22]. The diabetes-specific tool PAID was previously used in similar patient settings as well [67].

The results of this study add to the limited amount of literature on the differences in HRQOL determinants between age-matched older T1D and T2D patients and demonstrate that this is an important and understudied topic in diabetes. Incorporating and addressing the HRQOL perspective in diabetes care and research is imperative, especially for the elderly. In this subpopulation, a patient-centered approach to diabetes is crucial because of the prominent interindividual differences [65,88]. All care providers should include queries about self-rated health in routine care. Moreover, psychosocial care should be integrated with patient-centered medical care to optimize healthcare outcomes [66,89].

## 6. Conclusions

In the present study, HRQOL and self-rated mental health in older adults with diabetes were studied since T1D and T2D are known to significantly impact HRQOL and psychological well-being. Our study demonstrated some specificities in a subgroup of older adults such that most HRQOL domains were not associated with the presence of diabetes complications or glycemic control. Consequently, this study argues for a separate measure of patient-reported outcomes in addition to glycemic control. Furthermore, HRQOL assessment in the form of PROMs is gaining recognition as novel therapies and devices are increasingly introduced. Longitudinal research is also needed. Future research, especially incorporating randomized controlled trials, should be designed to capture PROMs since PROMs are central to treatment satisfaction, life and work productivity impairments, and cost-effectiveness studies [72,75]. Given that HRQOL measures are predictors of mortality in older adults [5], it is crucial to emphasize their value in targeted prevention efforts. Importantly, the most significant reduction in HRQOL was experienced by patients with higher BMIs, irrespective of diabetes type.

To conclude, the HRQOL data obtained in this study could be used in clinical settings for evidence-based education focused on specific subgroups of patients and in national healthcare policies targeting specific factors that influence outcomes, e.g., interventions designed to alleviate obesity.

## Figures and Tables

**Figure 1 healthcare-11-02154-f001:**
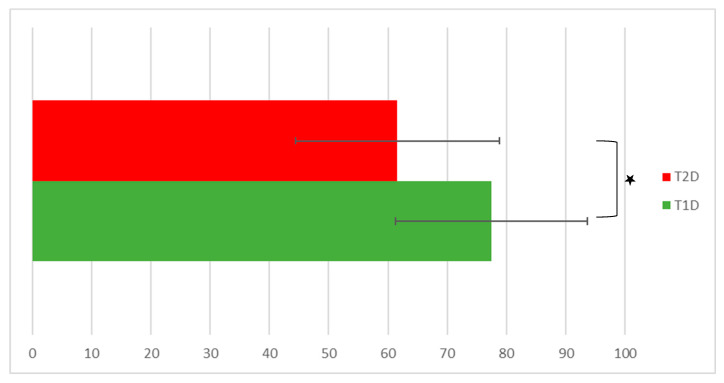
EQ-VAS scores according to diabetes type, indicating a worse overall impression of self-rated health in T2D participants. Legend: EQ-VAS—EuroQoL/Visual Analog Scale, T1D—type 1 diabetes patients, and T2D—type 2 diabetes patients. Values are presented as means ± SD. * Denotes statistical significance at *p* < 0.05.

**Figure 2 healthcare-11-02154-f002:**
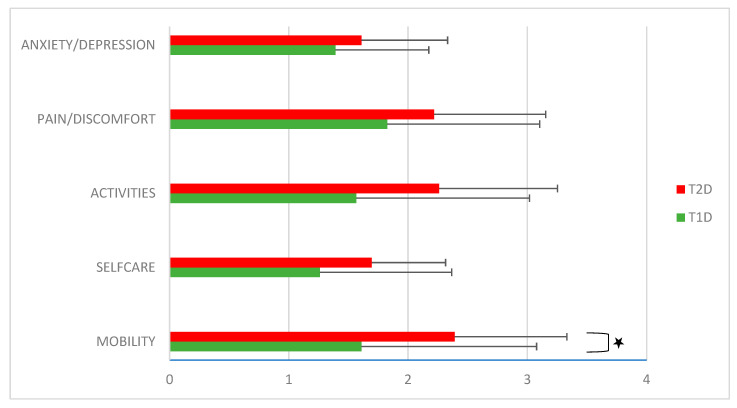
EQ-5D scores according to diabetes type, indicating more problems with mobility in T2D patients. Legend: EQ-5D—EuroQoL-5 dimensions, T1D—type 1 diabetes patients, and T2D—type 2 diabetes patients. Values are presented as means ± SD. * Denotes statistical significance at *p* < 0.05.

**Figure 3 healthcare-11-02154-f003:**
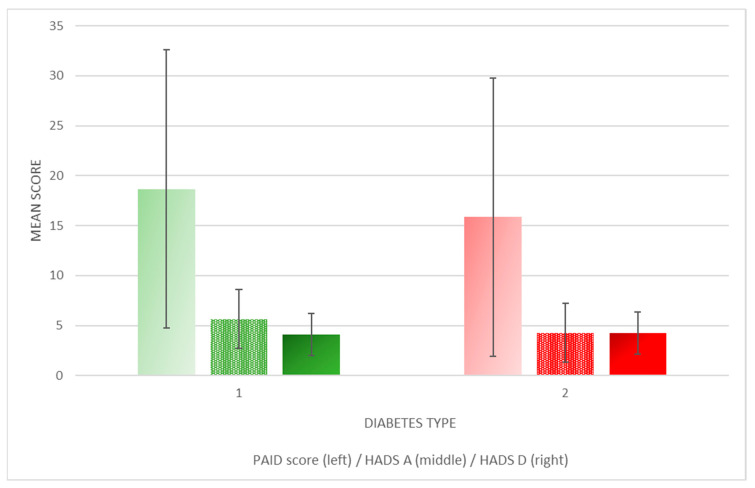
The Problem Areas in Diabetes (PAID) scale and Hospital Anxiety and Depression Scale (HADS) scores according to diabetes type did not significantly differ between T1D and T2D patients. Legend: HADS A—the Hospital Anxiety and Depression Scale Anxiety, HADS D—the Hospital Anxiety and Depression Scale Depression, and PAID—the Problem Areas in Diabetes scale.

**Table 1 healthcare-11-02154-t001:** The baseline characteristics of the included cohort of diabetic patients.

	T1D (*n* = 25)	T2D (*n* = 31)	*p*-Value
Age (years)	64.79 ± 5.89	72.26 ± 7.91	0.062
Female gender (%)	46	44	0.45
HbA1c (%; mmol/mol)	7.60 ± 1.37; 59.6 ± 14.0	8.12 ± 1.66; 65.2 ± 17.2	0.097
BMI (kg/m^2^)	27.68 ± 6.26	31.49 ± 7.49	0.047 *
Diabetes duration (years)	37.62 ± 15.7	20.45 ± 11.21	0.0001 *
Arterial hypertension prevalence (%)	80	88.5	0.38
Presence of microvascular complications (%)	92	76	0.14
Presence of macrovascular complications (%)	20.8	37.5	0.21

Legend: T1D—type 1 diabetes patients, T2D—type 2 diabetes patients, and BMI—body mass index. Values are presented as means ± SD. * Denotes statistical significance at *p* < 0.05.

**Table 2 healthcare-11-02154-t002:** Eight health dimensions of Medical Outcomes Short Form-36 (SF-36) according to diabetes type.

SF-36 Health Dimensions	T1D	T2D	*p*-Value
SF-PF	79.2 ± 20.3	67.6 ± 27.4	0.2
SF-LPH	75.0 ± 28.8	71.25 ± 29.5	0.6
SF-LEH	74.6 ± 25.6	83.33 ± 26.0	0.3
SF-V	62.5 ± 18.9	57.9 ± 19.5	0.5
SF-MH	76.0 ± 15.8	70.6 ± 16.9	0.5
SF-SF	78.4 ± 21.5	80.5 ± 16.0	0.6
SF-BP	63.1 ± 26.8	68.1 ± 27.4	0.6
SF-GH	58.6 ± 20.8	46.8 ± 20.1	0.048 *

Legend: physical functioning (PF), role limitation due to physical functioning (LPH), role limitation due to emotional health (LEH), vitality (V), mental health (MH), social functioning (SF), bodily pain (BP), and general health perception (GH). T1D: type 1 diabetes patients; T2D: type 2 diabetes patients. Values are presented as mean ± SD. * Denotes statistical significance at *p* < 0.05.

**Table 3 healthcare-11-02154-t003:** Predictors that affected respondents’ self-rated health scores for SF-36 domains and EQ-5D-VAS.

Predictor Outcome	Sex	Age	Duration	Diabetes Type	HbA1c	MicroV	MacroV	BMI
SF-PF	NS	NS	NS	NS	NS	NS	NS	–0.465 (0.002)
SF-BP	NS	NS	NS	NS	NS	NS	NS	−0.369 (0.043)
SF-V	NS	NS	NS	NS	NS	NS	NS	−0.417 (0.004)
SF-GH	NS	NS	NS	NS	NS	NS	NS	−0.346 (0.018)
EQ-5D VAS	NS	NS	NS	−0.451 (0.005)	NS	NS	NS	NS

The number in each of the brackets denotes the *p*-value; the estimated standardized regression coefficients Beta for the model-predicted values are listed numerically only where they contributed significantly to the model. Legend: SF—Short Form-36, PF: Physical functioning, SF-V: vitality, SF-BP: bodily pain, HADS A: the Hospital Anxiety and Depression Scale Anxiety, HADS D: the Hospital Anxiety and Depression Scale Depression, PAID: the Problem Areas in Diabetes scale. MicroV: microvascular complications; MacroV: macrovascular complications.

## Data Availability

The datasets generated during and/or analyzed during the current study are available from the corresponding author upon reasonable request.

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
