# Peer review of "Health-Related Quality of Life Assessment in Older Patients with Type 1 and Type 2 Diabetes"

_healthcare, 2023, doi:10.3390/healthcare11152154_

Round 1
Reviewer 1 Report
The work deals in a general way with different aspects of Type 1 and 2 Diabetes in older patients, knowing and comparing their health-related quality of life.
As the main drawback, I consider that the total sample is small, which means that the sample of each of the groups, Type 1 Diabetes and Type 2 Diabetes, also have quite small samples.
As a suggestion to the authors, I suggest reviewing Table 1. Homogenize the content of the table, for example, or put the % symbol in the first column or in the cells, but not sometimes in the first column and sometimes not in the cells. In Table 1 there is an error, when talking about HbA1c variable it cannot be % and then measured as a mean with a standard deviation.
In general, the work gives a broad vision of what the two types of diabetes represent in the lives of people over 65 years of age.
Author Response
The manuscript has been extensively revised for style, grammar, and content, with changes highlighted in Yellow and with the Track Changes function in the manuscript file. To improve the quality of the existing literature review, additional references were added.
Table 1 and Figure 3 have been transformed and hopefully improved in terms of clarity. We are thankful to the Reviewer for this remark. Since we acknowledge that the Graphical presentation adds to the recognition of the Article, we hope that the Graphical Abstract we submitted will be added to the body of the Manuscript and hopefully make it more appealing, if accepted for publication.
We would like to thank our reviewer for the constructive criticism and for the time dedicated to analyzing the manuscript. The responses to reviewers’ specific comments are attached below.

Reviewer 2 Report
Although this paper has some notes of interest, the quality of written english is very poor. Therefore, I definitely recommend a deep proofreading of mother tongue reviewer.
very scarse
Author Response
The manuscript has been extensively revised for style, grammar, and content, with changes highlighted in Yellow and with the Track Changes function in the manuscript file.
According to the Reviewer's suggestion, The MDPI English editing services were used for the entire manuscript.
To improve the quality of the existing literature review, additional references were added. Since we acknowledge that the Graphical presentation adds to the recognition and citing of the Article, we hope that the Graphical Abstract we submitted will be added to the body of the Manuscript and hopefully make it more appealing, if accepted for publication.
We would like to thank our reviewer for the constructive criticism and for the time dedicated to analyzing the manuscript.
Reviewer 3 Report
Dear Authors,
you have presented very interesting topic (but number of participants is very small, while you give information that diabetes is a common and widely present chronic disease in older adults), while I have some comments about it.
Section 2 should be Materials and Methods not only Methods
Section Methods - please add clearer inclusion and exclusion criteria of your study group - it will help Readers
p. 3, line 97-98 - please add references for two reaserch tools
subsections with Questionnaires - please add references not next to the name of Questionnaires but at the end of they descriptions (for example reference from line 100 put in line 107)
Section Results - first two sentence - information about number of participants - should be in section Materials and Methods
p. 8 line 229-231- this information should be before the table 3
Section Conclusion - in my opinin you should edit this section - please add clear conclusion from your own study and prosposal of practical implications form your study for example the need of future research in this area
Please folow the Instruction for Authors in context due to Author Contributions, Funding, Institutional Review Board Statement, Informed Consent Statement, Data Availability Statement, Acknowledgments, Conflicts of Interest
Author Response
The manuscript has been extensively revised for style, grammar, and content, with changes highlighted in Yellow and with the Track Changes function in the manuscript file. To improve the quality of the existing literature review, additional references were added.
Table 1 and Figure 3 have been transformed and hopefully improved in terms of clarity. We are thankful to the Reviewer for this remark. Since we acknowledge that the Graphical presentation adds to the recognition of the Article, we hope that the Graphical Abstract we submitted will be added to the body of the Manuscript and hopefully make it more appealing, if accepted for publication.
We would like to thank our reviewer for the constructive criticism and the time dedicated to analyzing the manuscript. The responses to reviewers’ specific comments are attached below.

Round 2
Reviewer 2 Report
Accepted
Author Response
Dear Reviewer,
The manuscript has been extensively revised for style, grammar, and content, as suggested.
We would like to thank our reviewers and the Editor for the constructive criticism and for their time dedicated to analyzing the manuscript. We believe their suggestions improved the quality of the paper.